# Association between Levocarnitine Treatment and the Change in Knee Extensor Strength in Patients Undergoing Hemodialysis: A Post-Hoc Analysis of the Osaka Dialysis Complication Study (ODCS)

**DOI:** 10.3390/nu14020343

**Published:** 2022-01-14

**Authors:** Shota Matsufuji, Tetsuo Shoji, Suhye Lee, Masao Yamaguchi, Mari Nishimura, Yoshihiro Tsujimoto, Shinya Nakatani, Tomoaki Morioka, Katsuhito Mori, Masanori Emoto

**Affiliations:** 1Division of Rehabilitation, Inoue Hospital, Enoki-cho Suita, Osaka 564-0053, Japan; matsufuji.shouta@aijinkai-group.com (S.M.); ri.suhe@aijinkai-group.com (S.L.); yamaguchi.masao@aijinkai-group.com (M.Y.); nishimura.mari@aijinkai-group.com (M.N.); 2Department of Vascular Medicine, Osaka City University Graduate School of Medicine, 1-4-3, Asahi-machi, Abeno-ku, Osaka 545-8585, Japan; 3Vascular Science Center for Translational Research, Osaka City University Graduate School of Medicine, Asahi-machi, Abeno-ku, Osaka 545-8585, Japan; memoto@med.osaka-cu.ac.jp; 4Division of Internal Medicine, Inoue Hospital, Enoki-cho Suita, Osaka 564-0053, Japan; tujimoto.yoshihiro@aijinkai-group.com; 5Department of Metabolism, Endocrinology and Molecular Medicine, Osaka City University Graduate School of Medicine, Asahi-machi, Abeno-ku, Osaka 545-8585, Japan; m2026719@med.osaka-cu.ac.jp (S.N.); m-tomo@med.osaka-cu.ac.jp (T.M.); 6Department of Nephrology, Osaka City University Graduate School of Medicine, Asahi-machi, Abeno-ku, Osaka 545-8585, Japan; ktmori@med.osaka-cu.ac.jp

**Keywords:** muscle strength, hemodialysis, carnitine deficiency, sarcopenia, frailty

## Abstract

Carnitine deficiency is prevalent in patients undergoing hemodialysis, and it could result in lowered muscle strength. So far, the effect of treatment with levocarnitine on lower limb muscle strength has not been well described. This observational study examined the association between treatment with levocarnitine with the change in knee extensor strength (KES) in hemodialysis patients. Eligible patients were selected from the participants enrolled in a prospective cohort study for whom muscle strength was measured annually. We identified 104 eligible patients for this analysis. During the one-year period between 2014 to 2015, 67 patients were treated with intravenous levocarnitine (1000 mg per shot, thrice weekly), whereas 37 patients were not. The change in KES was significantly higher (*p* = 0.01) in the carnitine group [0.02 (0.01–0.04) kgf/kg] as compared to the non-carnitine group [−0.02 (−0.04 to 0.01) kgf/kg]. Multivariable-adjusted regression analysis showed the positive association between the change in KES and the treatment with levocarnitine remained significant after adjustment for the baseline KES and other potential confounders. Thus, treatment with intravenous levocarnitine was independently and positively associated with the change in KES among hemodialysis patients. Further clinical trials are needed to provide more solid evidence.

## 1. Introduction

A large number of hemodialysis patients suffer from sarcopenia and decreased muscle strength which are closely related to impaired activity of daily living (ADL) [1] and high mortality rate [2,3,4]. Based on the diagnostic criteria of sarcopenia [5], strength of handgrip and lower extremities, in addition to muscle mass, are essential components of sarcopenia. Effective prevention and treatment for these muscle problems would result in improvement of their ADL and healthy longevity.

Sarcopenia and lowered muscle strength in hemodialysis patients are associated with age, body mass index (BMI), diabetes mellitus, and inflammation [4,6,7]. In addition to insufficient intake of protein and energy, the mechanisms for sarcopenia in kidney failure may include metabolic acidosis, inflammation, oxidative stress, decreased anabolic hormones, and others [8,9].

Carnitine deficiency may contribute to sarcopenia, lowered muscle strength, decreased cardiac function, anemia, and other complications of hemodialysis patients [10,11]. The mechanism by which carnitine deficiency results in decreased muscle strength is classically explained by the role of carnitine in energy metabolism. Namely, carnitine is present abundantly in myocardium and skeletal muscle, playing an essential role in energy production in mitochondria via ß-oxidation of long chain fatty acids. Carnitine is needed for the transport of fatty acids from cytoplasm into mitochondria [10,12], where long chain fatty acids undergo ß-oxidation for energy generation. In addition to the role in fatty acid ß-oxidation, carnitine is known to exert other potentially protective effects against muscle wasting [13]: the effect on insulin-like growth factor 1 (IGF-1) regulating the synthesis and degradation of body proteins, the effect on cytokines linking to inflammation, the effect on caspase 3 resulting in proteolysis and myonuclear apoptosis, the effect on oxidative stress, and the effect on mitochondrial dysfunction mediated by peroxisome proliferator-activated receptor-gamma coactivator (PGC)-1α. Induction of autophagy by carnitine also restores mitochondrial dysfunction in mice [14]. All these effects of carnitine are potentially protective against decreased muscle mass and strength. Hemodialysis patients have lower levels of plasma and muscle free carnitine than the non-dialyzed population [15]. Carnitine deficiency in hemodialysis patients is known to be caused by insufficient dietary intake, decreased carnitine synthesis in the body, and loss to dialysis fluid [16]. Previous studies in hemodialysis patients reported that treatment with levocarnitine improved cardiac function and exercise tolerance [17,18,19]. Regarding its potential effects on skeletal muscle in patients treated with hemodialysis, some studies reported improvement of muscle spasm following levocarnitine treatment [17,20,21,22]. However, information is limited regarding its effects on muscle strength [23,24,25]. Particularly, no study is available in the literature that examined the effect of levocarnitine on muscle strength of lower extremity, which is important for independence of ADL [26].

In the present study, we examined whether treatment with levocarnitine was associated with the changes of knee extension strength in hemodialysis patients.

## 2. Materials and Methods

### 2.1. Study Design

This was a single center, observational study in hemodialysis patients including those with and without intravenous levocarnitine injection as a part of usual care. The exposure was the presence versus absence of treatment with levocarnitine. The primary outcome was the change in knee extensor strength in one year (2014 to 2015). The secondary outcome was the change in handgrip strength in the same period. To address the question whether the observed association between the exposure and the primary outcome was attributable to the levocarnitine treatment, similar analyses were conducted using the data obtained in the 2013–2014 period before levocarnitine treatment was started.

### 2.2. Participants

We selected the patients of this analysis from the hemodialysis patients treated at the out-patient clinic of the dialysis center of Inoue Hospital who had been enrolled in a multicenter, prospective cohort study (The Osaka Dialysis Complication Study, ODCS) [27].

A total of 1696 patients from 17 dialysis facilities were enrolled in the ODCS in 2012, and the cohort was followed-up for five years until 2017 to examine the factors associated with the changes in cognitive function and other clinical outcomes. As a part of the ODCS, the participants at Inoue Hospital were annually checked-up by physical trainers for their ADL using the Barthel Index and muscle strength. During the follow-up period of the ODCS, intravenous levocarnitine was approved for clinical use in Japan in 2014, and it was administered to the ODCS participants if indicated.

The eligible patients for this analysis were determined by the following selection and exclusion criteria. The selection criteria were (1) patients treated at the out-patient dialysis center of Inoue Hospital who were enrolled in the ODCS with written informed consent in 2012, (2) patients whose ADL was independent in 2013, (3) patients in whom both knee extensor strength and handgrip strength were measured serially in 2013, 2014, and 2015, and (4) patients who were treated or not treated with intravenous levocarnitine. The exclusion criteria were (1) patients who experienced death, transfer to another medical institution, or switching to other treatment modality than hemodialysis, (2) patients who had orthopedical surgery during the period of 2013 through 2015, and (3) patients who started the treatment with levocarnitine, but the treatment was not continued for one year. The exclusion criterion (3) was set because treatment with levocarnitine was discontinued in some patients according to the instructions on the package insert as described below.

### 2.3. Treatment with Levocarnitine

In Japan, levocarnitine was approved for clinical use and indicated for patients with carnitine deficiency diagnosed by laboratory tests. We diagnosed carnitine deficiency when the serum free carnitine level was lower than 40 μmol/L, according to Cascaiani et al. [28]. As a part of usual care, treatment with levocarnitine was considered for patients with carnitine deficiency plus anemia resistant to erythropoiesis-stimulating agent (ESA) and/or muscle symptoms such as muscle weakness and muscle clamp. In previous studies with hemodialysis patients using intravenous levocarnitine, it was given at the end of each dialysis session at a dose of 10–40 mg levocarnitine per kg of body weight [14,16] or at a fixed dose of 2000 mg per shot [20]. In our practice during 2014 through 2015, one ampoule of levocarnitine solution (L-cartin^®^ injection 1000 mg syringe, Otsuka Pharmaceutical Co., Ltd. Tokyo, Japan) containing 1000 mg of levocarnitine and dilute hydrochloride as a pH adjusting agent was intravenously administered thrice weekly at the end of each hemodialysis session. This corresponded to 10.6–26.3 mg levocarnitine per kg of body weight. The treatment was discontinued if it was not effective for either anemia or muscle symptoms according to the instructions on the package insert of L-cartin^®^ injection. Although both oral and intravenous preparations were available for clinical use, only the intravenous preparation of levocarnitine was used in our hospital.

### 2.4. Measurement of Muscle Strength

Knee extensor strength was measured by a chair-shaped digital dynamometer for legs (Model TP-776S, Toyo Physical Co. Ltd., Fukuoka, Japan) in a sitting position with the hip and knee joints at 90° flexion as previously described [29]. We recorded two consecutive measurements of 5 s isometric strength for each side, and the higher value was used for analysis. The value in knee extensor strength was corrected for body mass by dividing it by body weight in kilograms of the patient. We used body weight after removing excessive body fluid by dialysis, so called “dry weight”, according to previous reports [29,30].

The handgrip strength was measured by a digital dynamometer (Grip D; Model T.K.K. 5101, Takei Scientific Instruments Co. Ltd., Niigata, Japan) in a standing position with the participant’s arm put down and the elbow joint extended [31,32]. For each side, two consecutive measurements were done, and the higher value was recorded. The highest value of the four measurements was used for analysis regardless of dominant or non-dominant hand.

### 2.5. Other Variables

We collected data on age, sex, BMI, duration of hemodialysis, underlying kidney disease (diabetic kidney disease or not), and history of cardiovascular disease (CVD) including coronary artery disease, stroke, and peripheral artery disease from medical records. In addition, we collected laboratory data including hemoglobin, serum albumin, C-reactive protein (CRP), total carnitine, free carnitine, acyl carnitine, and acyl/free carnitine ratio.

### 2.6. Statistical Analysis

First, the clinical characteristics in 2014 (baseline) were compared between the two groups. We summarized continuous variables as medians (interquartile ranges) and categorical variables as numbers (percentages). Comparison was done by the Mann–Whitney U-test and χ^2^ test, respectively. We used non-parametric statistics because some variables did not follow normal distribution.

For knee extensor strength, comparison between 2014 and 2015 was done by the Wilcoxon signed rank test. The change in knee extensor strength was calculated by subtracting the value in 2014 from the value in 2015. The between-group comparison of the change in knee extensor strength during the period was done by the Mann–Whitney U-test. Then, we examined the association between the change in knee extensor strength and the levocarnitine treatment using multivariable-adjusted regression analysis with adjustment for possible confounders including the baseline knee extensor strength, age, sex, duration of hemodialysis, diabetic kidney disease or not, and prior CVD. The residual was deemed to follow normal distribution.

We performed the following two additional analyses for the interpretation of the primary results. First, to assess whether the observed between-group difference in the period between 2014 to 2015 was attributable to levocarnitine treatment, we conducted similar analyses using the data of knee extensor strength in the period between 2013 and 2014 when none was treated with levocarnitine. Second, we analyzed the data of handgrip strength in place of knee extensor strength to address whether levocarnitine treatment was associated with the change in muscle strength other than the knee extensor.

These statistical calculations were performed using statistical software JMP 12 (SAS Institute Japan, Tokyo, Japan) for Windows personal computers. *p* < 0.05 by a two-sided test was considered statistically significant.

## 3. Results

### 3.1. Selection of Study Participants

Figure 1 shows the selection of study participants. We had 335 hemodialysis patients at Inoue Hospital who were enrolled in the ODCS in 2012. Among them, we identified 120 patients for whom muscle strength was serially measured in 2013, 2014, and 2015. Sixteen patients were excluded because of levocarnitine treatment was discontinued in three months. Finally, 104 patients were selected for this study. Among them, 67 patients were treated with levocarnitine from 2014 to 2015 (carnitine group), whereas 37 patients were not (non-carnitine group).

### 3.2. Clinical Characteristics of the Participants in 2014

Table 1 shows the clinical characteristics of the carnitine and the non-carnitine groups in the year of 2014 just before the levocarnitine treatment was started in the carnitine group. No significant difference was found in age, sex, duration of hemodialysis, diabetic kidney disease or not, prior CVD, or BMI. The carnitine group showed lower values of total, free and acyl carnitine than the non-carnitine group, because levocarnitine treatment was indicated for carnitine deficiency.

### 3.3. Changes in Knee Extensor Strength during the Period before Levocarnitine Treatment

Figure 2A shows the measurements of knee extensor strength and the changes during the 2013–2014 period before levocarnitine treatment was started. The knee extensor strength was decreased in both the carnitine group [0.42 (0.37–0.54) to 0.41 (0.34–0.48) kgf/kg, *p* = 0.004] and the non-carnitine group [0.49 (0.41–0.56) to 0.44 (0.38–0.56) kgf/kg, *p* = 0.004]. The change in knee extensor strength was not different (*p* = 0.41) between the carnitine group [−0.03 (−0.05 to 0.01) kgf/kg] and the non-carnitine group [−0.04 (−0.07 to 0.02) kgf/kg].

### 3.4. Changes in Knee Extensor Strength during the Period of Levocarnitine Treatment

Figure 2B shows the measurements of knee extensor strength and the changes during the 2014–2015 period when levocarnitine treatment was done in the carnitine group. The knee extensor strength was decreased in the non-carnitine group [0.44 (0.38–0.56) kgf/kg to 0.44 (0.34–0.52) kgf/kg, *p* = 0.04], whereas it was increased in the carnitine group [0.41 (0.34–0.48) kgf/kg to 0.43 (0.35–0.50) kgf/kg, *p* = 0.02]. The change in knee extensor strength was significantly higher (*p* = 0.01) in the carnitine group [0.02 (0.01–0.04) kgf/kg] as compared to the non-carnitine group [−0.02 (−0.04 to 0.01) kgf/kg].

### 3.5. Independent Association of Treatment with Levocarnitine with the Change in Knee Extensor Strength

We examined whether the observed between-group difference in the one-year change (from 2013 to 2014, or from 2014 to 2015) in knee extensor strength was attributable to the levocarnitine treatment by using the multivariable-adjusted linear regression model which included age, sex, duration of hemodialysis, diabetic kidney disease or not, prior CVD, and knee extensor strength at the beginning of the period as potential confounders (Table 2). In the analysis of date during the 2013–2014 period before levocarnitine treatment was done, the patient group was not associated with the change in knee extensor strength. In contrast, in the analysis of data during the 2014–2015 period when levocarnitine treatment was done in the carnitine group, the significantly greater change (increase) in knee extensor strength was associated with the carnitine group versus the non-carnitine group, independent of the possible confounders. A higher knee extensor strength at the beginning of the period was inversely associated with its change during the period. Other factors did not have significant associations with the change in knee extensor strength.

### 3.6. Independent Association of Treatment with Levocarnitine with the Change in Handgrip Strength

Figure 3A shows the changes in handgrip strength in the two groups during the 2013–2014 period before levocarnitine treatment was started. No significant change was observed in either group. Additionally, the change in handgrip strength was not significantly different between the two groups in this period.

Figure 3B shows the changes in handgrip strength in the two groups during the 2014–2015 period when levocarnitine treatment was done in the carnitine group. Despite the treatment, handgrip strength was decreased in the carnitine group, whereas no significant change was observed in the non-carnitine group. The between-group comparison showed no significant difference in the change in handgrip strength during this period.

## 4. Discussion

In this observational study, we showed the increase in knee extensor strength in patients who were treated with levocarnitine, whereas it was decreased in patients who were not treated. The association between the treatment with levocarnitine and the change in knee extensor strength was independent of potential confounders. Such an association between the treatment group and the change in knee extensor strength was not found before the levocarnitine treatment was started. In contrast, no significant association was found between the treatment with levocarnitine and the change in handgrip strength. These observational data indicate that treatment with levocarnitine was more preferentially associated with the change in knee extensor strength than the change in handgrip strength in patients undergoing hemodialysis.

Knee extensor strength was decreased regardless of the patient group in the period of 2013–2014 before levocarnitine was started. This finding in hemodialysis patients is consistent with the results in patient groups other than hemodialysis patients [33,34,35]. More importantly, our results clearly showed that treatment with levocarnitine was an independent factor associated with increased knee extensor strength in hemodialysis patients. Regarding the effect of levocarnitine on knee extensor strength in hemodialysis patients, Giovenali et al. [23] reported important findings. They conducted a one-arm 24-week trial in which 2 g of levocarnitine was given either intravenously at the end of hemodialysis or orally twice daily in 26 hemodialysis patients. Although their main purpose was to observe morphological changes in quadriceps muscle fibers by biopsies, they also showed the increase in knee extensor strength following levocarnitine treatment. The result of our study was consistent with their report [23]. In addition, although the study by Giovenali did not have a control group without treatment with levocarnitine, we were able to compare the change in knee extensor strength between the two groups with and without levocarnitine treatment.

In contrast to the significant association between levocarnitine treatment and the increase in knee extensor strength, we could not demonstrate a significant between-group difference in the change in handgrip strength during the period of 2014–2015 when levocarnitine was administered in the carnitine group. At the same time, the treatment with levocarnitine was not an independent factor associated with the change in handgrip strength during the same period. The findings of this observational study are consistent with previous reports of randomized controlled trials by Maruyama et al. [24] and by Yano et al. [25] conducted in hemodialysis patients in the fact that levocarnitine treatment did not increase handgrip strength. Moreover, no increase in handgrip strength was reported following levocarnitine treatment in an uncontrolled trial in nonrenal elderly individuals [36]. In addition to the increase in knee extensor strength, levocarnitine treatment was reported to increase the circumference and muscle volume of the thigh of hemodialysis patients [25]. Thus, the treatment with levocarnitine appears to have a more preferential association or effects on muscles related to knee extension than handgrip.

How can we explain the different associations of levocarnitine treatment with the changes in knee extensor strength and handgrip strength in this study? The different responsiveness to the treatment may be attributable to the difference in muscle fiber composition. As mentioned above, Giovenali et al. [23] performed repeated quadriceps muscle biopsies before and after levocarnitine treatment in hemodialysis patients and showed that atrophy of type 1 and type 2A muscle fibers was improved, whereas no improvement was found in type 2B muscle fibers, suggesting that levocarnitine treatment improved muscle fiber types that generate energy via fatty acid oxidation than glycogen. As compared to type 2B fiber, type 1 and 2A fibers contain more mitochondria in which energy is generated via ß-oxidation of long-chain fatty acids which need carnitine for their transport from cytosol into mitochondria [10,12]. Thus, we speculate that the compositional difference in muscle fibers with different metabolic characteristics may explain the preferential association of levocarnitine treatment with the increase in knee extensor strength than handgrip strength observed in this study. However, we could not find previous reports that compared muscle fiber composition between muscles responsible for knee extensor strength (such as quadriceps muscle) and handgrip strength (such as flexor digitorum superficialis, flexor digitorum profundus, and flexor pollicis longus). Thus, the above-mentioned explanation is a speculation which needs further study. Additionally, other potential mechanisms than fatty acid ß-oxidation cannot explain why handgrip strength did not increase following levocarnitine. Therefore, the mechanism is unknown for the preferential association of levocarnitine treatment with increased knee extensor strength.

There are several limitations in this study. First, because this study includes only hemodialysis patients in Japan, we are not sure that the results of this study are applicable to other populations. Second, because we excluded patients who discontinued the levocarnitine treatment in three months following the instruction on the package insert, such exclusion may affect the results. Third, the non-carnitine group served as a control group, but the characteristics of the two groups were different, unlike randomized controlled trials. To address this issue, statistical adjustment was done using multivariable-adjusted linear regression models. Fourth, although the change in knee extensor strength may increase physical activity [37,38], we did not examine such association with levocarnitine treatment. Fifth, because levocarnitine treatment was done only in patients with carnitine deficiency as evidenced by low serum free carnitine levels, it is unknown whether the same treatment is associated with muscle strength in those with normal serum carnitine levels. Sixth, we did not have repeated measurements of serum carnitine levels. Therefore, we are unable to show how much the supplementation increased serum carnitine levels in the carnitine group. Moreover, it is unknown whether serum carnitine levels stayed unchanged or decreased over time without supplementation in the non-carnitine group. Seventh, although we measured muscle strength, we did not perform measurement of muscle mass or diagnosis of the presence of sarcopenia. Eighth, there might be the effect of familiarization or learning effect in the increase of knee extension strength measurement. However, it is unlikely, because such effect, if any, should be cancelled out in the between-group comparison. In addition, the measurements were done at an interval of one year.

This study showed the association between carnitine treatment with increased knee extensor strength in patients with kidney failure treated with hemodialysis. Supplementation of levocarnitine and other carnitine derivatives are also known to improve muscle wasting signs and symptoms in various conditions such as neurofibromatosis type 1 [39] and sarcopenia in chronic liver disease [40]. Furthermore, carnitine supplementation also improves recovery and reduces fatigue following exercise in the general population [41]. These results warrant further research of levocarnitine as a treatment of sarcopenia and muscle wasting in various populations.

In conclusion, this observational study in the carnitine and non-carnitine groups clearly demonstrated that treatment with levocarnitine was associated with an increase in knee extensor strength among hemodialysis patients in the carnitine group. In addition, levocarnitine treatment was more preferentially associated with the change in knee extensor strength than the change in handgrip strength. These observations, along with previous results by clinical trials, support the potential benefit on knee extensor strength of levocarnitine treatment in hemodialysis patients with carnitine deficiency. Further well-designed trials are needed to draw more solid conclusions. This study could be a good starting point for such clinical trials.

## Figures and Tables

**Figure 1 nutrients-14-00343-f001:**
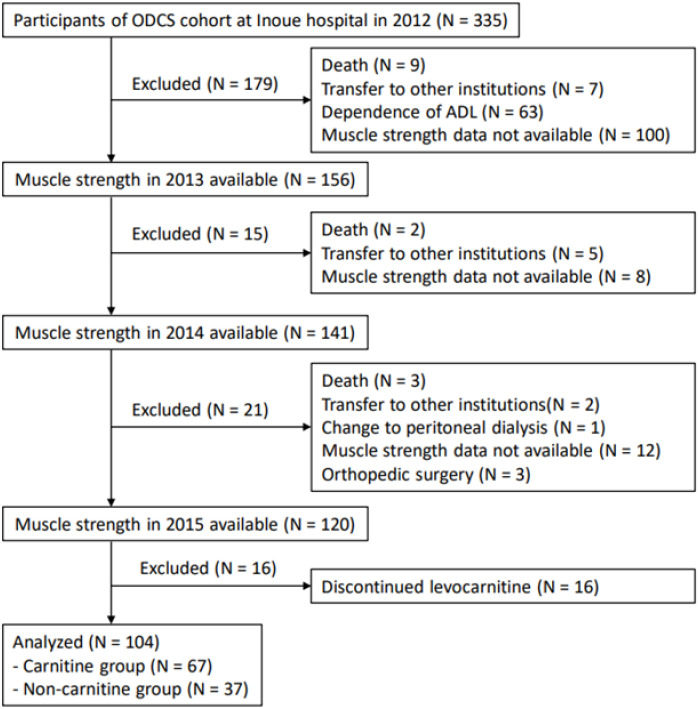
Selection of study participants. The patients analyzed in this study were selected from the 335 participants of the Osaka Dialysis Complication Study treated at Inoue Hospital. According to the inclusion and exclusion criteria, 104 eligible patients were included. Abbreviation: ODCS, Osaka Dialysis Complication Study; ADL, activity of daily living.

**Figure 2 nutrients-14-00343-f002:**
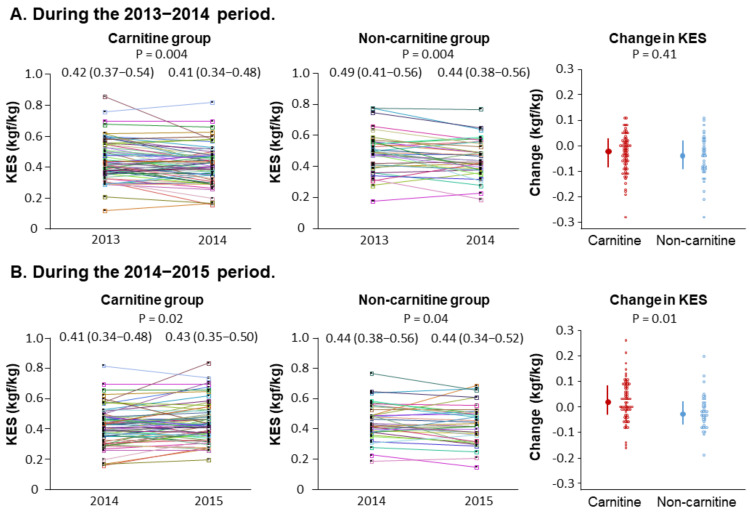
Changes in knee extensor strength. The graphs show individual measurements of knee extensor strength in the carnitine and non-carnitine groups and the between-group comparison of the one-year changes during the 2013–2014 period (**A**) and during the 2014–2015 period (**B**). In the carnitine group, treatment with levocarnitine was done only in the period from 2014 and 2015. The error bars represent interquartile ranges. Abbreviation: KES, knee extensor strength.

**Figure 3 nutrients-14-00343-f003:**
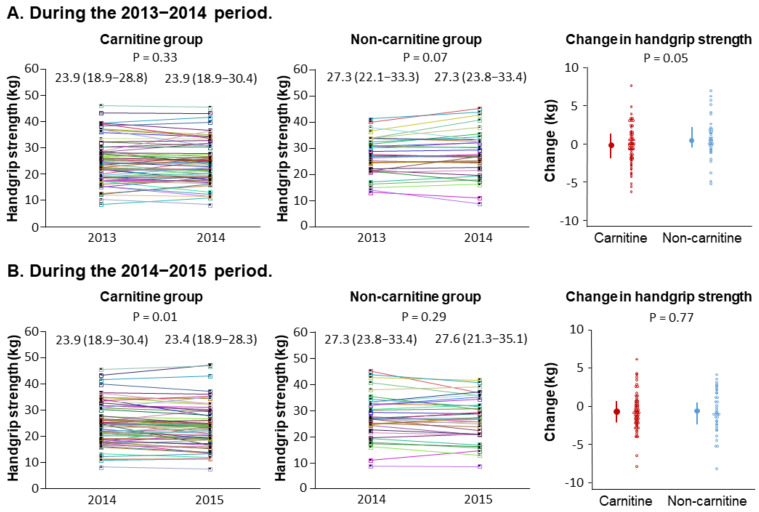
Changes in handgrip strength. The graphs show individual measurements of handgrip strength in the carnitine and non-carnitine groups and the between-group comparison of the one-year changes during the 2013–2014 period (**A**) and during the 2014–2015 period (**B**). In the carnitine group, treatment with levocarnitine was done only in the period from 2014 and 2015. The error bars represent interquartile ranges.

**Table 1 nutrients-14-00343-t001:** Characteristics of participants.

	Total Participants (*N* = 104)	Carnitine Group (*N* = 67)	Non-Carnitine Group (*N* = 37)	*p* Value
Age (year)	65 (58–71)	64 (56–71)	65 (58–72)	0.66
Female sex [*N* (%)]	35 (33.6%)	25 (37.3%)	10 (27.0%)	0.28
Duration of hemodialysis (year)	10 (4–17)	11 (5–20)	7 (4–14)	0.14
Diabetes mellitus [*N* (%)]	29, (27.9%)	22 (32.8%)	7 (18.9%)	0.12
Prior cardiovascular disease [*N* (%)]	22 (21.1%)	16 (23.9%)	6 (16.2%)	0.35
Body weight (kg)	56.0 (50.0–63.1)	55.5 (50.0–63.0)	57.4 (49.3–64.0)	0.51
Body mass index (kg/m^2^)	21.3 (19.3–23.0)	21.0 (19.2–22.6)	21.3 (18.9–23.6)	0.77
Hemoglobin (g/dL)	10.9 (10.4–11.6)	10.8 (10.4–11.5)	11.2 (10.5–11.6)	0.39
Serum albumin (g/dL)	3.9 (3.8–4.1)	3.9 (3.8–4.1)	3.9 (3.7–4.1)	0.59
C-reactive protein (mg/dL)	0.10 (0.10–0.19)	0.10 (0.10–0.19)	0.10 (0.10–0.22)	0.71
Total carnitine (μmol/L)	37.3 (31.5–46.8)	36.5 (31.3–45.7)	76.1 (50.4–122.0)	0.004
Free carnitine (μmol/L)	20.7 (16.0–24.9)	20.4 (15.8–23.1)	44.2 (31.4–72.2)	0.001
Acyl carnitine (μmol/L)	17.1 (14.2–21.5)	17.0 (14.3–20.1)	31.9 (19.0–49.8)	0.02
Acy/free carnitine ratio	0.86 (0.71–1.02)	0.87 (0.73–1.03)	0.61 (0.56–0.81)	0.03
Handgrip strength (kg)	25.4 (19.7–31.9)	23.9 (18.7–30.4)	27.3 (23.8–33.4)	0.01
Knee extensor strength (kgf/kg)	0.43 (0.36–0.49)	0.41 (0.33–0.48)	0.44 (0.38–0.56)	0.09

The table indicates numbers, percentages, or median (interquartile range) values. *p* values were by Mann–Whitney U test or χ^2^ test.

**Table 2 nutrients-14-00343-t002:** Multivariable-adjusted linear regression analysis of factors associated with the change in knee extensor strength.

Exposure Variables	Outcome Variables					
	Change in Knee Extensor Strength from 2013 to 2014			Change in Knee Extensor Strength from 2014 to 2015		
	Coefficient (95% CI)	*p* Value	Std. Coefficient	Coefficient (95% CI)	*p* Value	Std. Coefficient
Age (year)	0.0004 (−0.003 to 0.004)	0.81	0.03	−0.001 (−0.003 to 0.002)	0.06	−0.19
Sex (female = 0, male = 1)	−0.01 (−0.05 to 0.02)	0.48	−0.07	−0.01 (−0.024 to 0.008)	0.31	−0.10
Duration of hemodialysis (year)	0.002 (−0.002 to 0.007)	0.23	0.13	−0.001 (−0.003 to 0.001)	0.32	−0.10
Diabetes mellitus (yes = 1, no = 0)	0.01 (−0.03 to 0.05)	0.53	0.07	−0.006 (−0.024 to 0.011)	0.49	−0.07
Prior cardiovascular disease (yes = 1, no = 0)	−0.004 (−0.05 to 0.04)	0.86	−0.02	0.02 (−0.003 to 0.03)	0.10	0.15
Knee extensor strength at the beginning of the period (kgf/kg)	–0.44 (−0.71 to −0.17)	0.002	−0.35	−0.19 (−0.32 to −0.07)	0.003	−0.31
Group (Carnitine = 1, Non-carnitine = 0)	−0.01 (−0.05 to 0.02)	0.43	−0.08	0.02 (0.002 to 0.03)	0.03	0.23

Associations of patient group with one-year changes in knee extensor strength in the period before and during treatment with carnitine were examined with a multivariable-adjusted linear regression model. Adjustment was done for age, sex, hemodialysis duration, diabetes mellitus, prior cardiovascular disease, and knee extensor strength at the beginning of the period. In the carnitine group, treatment with levocarnitine was done only in the period from 2014 and 2015. The coefficients in the table are the point estimates (95% CI). Abbreviation: CI, confidence interval; Std. coefficient, standardized coefficient.

## Data Availability

The data will be available upon reasonable request to the corresponding author.

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
