# Peer review of "Association between Levocarnitine Treatment and the Change in Knee Extensor Strength in Patients Undergoing Hemodialysis: A Post-Hoc Analysis of the Osaka Dialysis Complication Study (ODCS)"

_nutrients, 2022, doi:10.3390/nu14020343_

Round 1
Reviewer 1 Report
The authors address the question whether levocarnitine supplementation has an impact on muscle strength in chronic hemodialysis patients in a post-hoc analysis of an initially prospective, single center observational study.
The question is very, the findings are potentially interesting, as sarcopenia is an important topic for hemodialysis patients and ways to avoid it would be very valuable.
However, the data are not very strong and the groups compared are not "equal" as they differed in therapy indication (only low carnitine patients were treated).
Few minor issues:
- As only patients were treated who had low carnitine levels, I would find it appropriate to write that accordingly.
Levocarnitine supplementation in patients with low carnitine levels had the effect on knee extensor strength, not generally – it was not randomized, it was observational and the groups differed in carnitine level as this was how they were assigned to treatment or not.
This is also one further limitation of the study – we do not know if patients with normal carnitine levels would have gained strength. (Baseline differences are ~2fold.) - At baseline, Carnitine levels are about twice as high in the non-treated group than in the treatment group (table 1).
I would like to see the carnitine levels measured after one year of levocarnitine supplementation: Do you have these data?
There we would learn 1.) how well supplementation worked and 2.) if the patients without supplementation remained stable or lost carnitine over time. - Figure 2: could you also include all data points in your graph showing changes in KES? Maybe as single data points in a grey shade in the background?
While average plus IQR is nice, it would be nicer and also informative to see all data points (plus average/IQR). - Please go over the manuscript with a native speaker or someone close to. (Please excuse this point – if I had to publish in Japanese, my text would be unreadable. Your manuscript is very readable, but some words/some sentences would be easier to understand for an international medical audience if the text was reviewed for English language.) Examples to correct would be “thigh” instead of “thy” (page 9, muscle volume of the …) among others. “Treated and untreated” in the abstract. Etc.
- On page 10, the authors speculate about why knee extensor strength is changed but hand grip is not, and then they argue with muscle fibre composition of quadriceps versus biceps. To my knowledge, biceps muscle is not involved in hand grip.
Do you have data on more distal muscle groups that strengthen your point?
Otherwise I would leave this speculation out. While carnitine is important for mitochondria and thereby Fatty acid oxidation, you do not have any data point on muscle metabolism in either muscle group (involved in knee extension or hand grip) in your observational post-hoc analysis. - I do very much agree with the last point of the authors that further randomized, controlled and prospective studies are needed to address if and how levocarnitine improves muscle strength in hemodialysis patients. Maybe there could be a little more emphasis on this point.
This study could be a good starting point for a randomized, controlled, prospective trial.
Reviewer 2 Report
I read with great interest the manuscript entitled “Association between levocarnitine treatment and the change in 2 knee extensor strength in patients undergoing hemodialysis: A 3 post-hoc analysis of the Osaka Dialysis Complication Study 4 (ODCS)
” which I found very interesting and relevant for the field. However, I found some issues that are difficult to retrieve:
- Introduction lacks continuity and strength regarding the topic.
- No association of the knee extension strength and grip strength with ALD, for example, is missing
- Regarding participants, no criteria for sarcopenias was assessed, as the authors assumed that hemodialysis patients are sarcopenic, as they state in the introduction.
- Levocarnitine type is not described (i.e., Hydrochloride). Also, the reared assumed that the doses stated on lines 111-113 then 3000mg are administrated with no previous amounts referenced.
- Authors described muscle strength measurement as normalizing by “dry weight,” but this is not accurate in humans. The normalization needs to be assessed by body or lean mass, but “dry o wet” weight is assessed for other measurements. ( https://doi.org/10.3389/fphys.2021.767941)
- The familiarization of all tests is also not described. All these tests possess a learning curve; this is a critical factor in determining the real difference in the expected changes.
- It is unclear whether HG strength was assessed on the dominant or non-dominant hand, the same for isometric strength regarding legs.
- Regarding statistical analysis, normalization of the data was not assessed, as the authors described the non-parametrical test as a default; this needs to be explained.
- On lines 114-116 delta changes, would it be more accurate to describe the subtraction of the years on the strength data
- The author states on lines 42-44 that hemodialysis patients are associated with inflammation, but values at leas of CPR are not altered between changes on this cohort. Hence, it seems that levocarnitine does not affect this parameter.
- No data on how much the patients with supplementation increase their Carnitine values after all the treatment.
- The authors explain that non-treated but “normal” carnitine levels lose KES? As stated, this deficiency would be causing a loss in strength.
- The authors concluded that “this observational study in the carnitine and non-carnitine groups demonstrated that treatment with levocarnitine was associated with an increase in knee extensor strength among hemodialysis patients” However the data do not support this assumption with the current methodological issues.
Round 2
Reviewer 2 Report
Thanks to the authors for considering all the commentaries previously assessed that reconsider my previous decision.
Regarding the new version, I still have some commentaries that are described below:
-Still, the introduction seems a little vague regarding the form described. By this, I mean that authors only add "lowered muscle strength" as a word in 2 lines, but the continuity of the introduction needs to be linked to their primary focus. (i.e., No Handgrip as a marker of strength is named). Also, sarcopenia is described as a loss of muscle mass and strength, not only in lower body mass. Please reconsider re-write this.
Line 49 states that carnitine deficiency may contribute to sarcopenia, but no possible mechanism is named or described. It will be helpful to cite at least some studies that explained this in particular and not only by b oxidation.
-On lines 135-135, please add a reference to the adjustment of these particular methods removing body fluids.
On lines 395-299, it will be essential to describe a possible mechanism of carnitine, as I mentioned earlier on the strength response. Is there something that may be on mice? Also will be important as sarcopenia is associated with weakness in these particular patients; why do carnitine levels decrease over the years? Coul be important to other muscle weakness diseases?
-As described, as no mechanism of the possible effect of carnitine on strength, lines 318-337 leave the reader with a notion that this is associated with fat utilization, which technically makes no sense; please reorder this to try to explain metabolism linked to this better function.
